# Neoteric Algorithm Using Cell Population Data (VCS Parameters) as a Rapid Screening Tool for Haematological Disorders

**DOI:** 10.3390/diagnostics11091652

**Published:** 2021-09-09

**Authors:** Angeli Ambayya, Jameela Sathar, Rosline Hassan

**Affiliations:** 1Clinical Haematology Referral Laboratory, Haematology Department, Hospital Ampang, Selangor 68000, Malaysia; jsathar@hotmail.com; 2Department Haematology, School of Medical Sciences, Universiti Sains Malaysia, Kelantan 15200, Malaysia

**Keywords:** cell population data, algorithm, neoplastic, VCS parameters

## Abstract

Hitherto, there has been no comprehensive study on the usefulness of cell population data (CPD) parameters as a screening tool in the discrimination of non-neoplastic and neoplastic haematological disorders. Hence, we aimed to develop an algorithm derived from CPD parameters to enable robust screening of neoplastic from non-neoplastic samples and subsequently to aid in differentiating various neoplastic haematological disorders. In this study, the CPD parameters from 245 subtypes of leukaemia and lymphoma were compared against 1103 non-neoplastic cases, and those CPD parameters that were vigorous discriminants were selected for algorithm development. We devised a novel algorithm: [(SD-V-NE*MN-UMALS-LY*SD-AL2-MO)/MN-C-NE] to distinguish neoplastic from non-neoplastic cases. Following that, the single parameter MN-AL2-NE was used as a discriminant to rule out reactive cases from neoplastic cases. We then assessed CPD parameters that were useful in delineating leukaemia subtypes as follows: AML (SD-MALS-NE and SD-UMALS-NE), APL (MN-V-NE and SD-V-MO), ALL (MN-MALS-NE and MN-LMALS-NE) and CLL (SD-C-MO). Prospective studies were carried out to validate the algorithm and single parameter, MN-AL2-NE. We propose these CPD parameter-based discriminant strategies to be adopted as an initial screening and flagging system in the preliminary evaluation of leukocyte morphology.

## 1. Introduction

Haematological malignancies typically arises from two major blood cell lineages: lymphoid and myeloid cells [1], which can progress into different subtypes of leukaemia, lymphoma, and multiple myeloma [2]. Haematological malignancies are the fourth most frequently diagnosed cancer around the world, with an incidence rate of 9% of all cancers [3]. For decades, the diagnosis of haematological malignancies has relied on morphologic examinations of the subject’s blood or bone marrow aspirates [4], and by immunophenotype profiling using flow cytometry techniques [5]. The advent of molecular genetics and genomic technologies, such as polymerase chain reaction (PCR) and sequencing-based methods, enabled deeper insight into the diagnosis and prognosis of various haematological malignancies [6]. Due to restricted number of trained personnel, the lack of sophisticated laboratory diagnostic equipment and the need to refer to the tertiary hospitals in rural healthcare settings [7], the diagnostic workups are arduous, leading to delays in initiating patients’ treatment protocols and, in some cases, resulting in untimely death due to these limitations.

The advent of state-of-the-art full blood count (FBC) analysers that are equipped with advanced parameters, such as cell population data (CPD) information, provides useful information for an initial screening and flagging system, enabling timely patient work up and diagnosis in settings with limited resources. Newer generation FBC analysers, such as Unicel DxH 800, are equipped to perform complex analysis of haematological cellular structures with similar working principles to flow cytometry by using laser to distinguish the volume (V), conductivity (C), and size (S) of blood cells, also known as the VCS parameters [8,9]. With similar data presentation to flow cytometry, this newer generation of FBC analysers generates cell population data (CPD) which details the different clusters of cell populations, such as leukocytes and erythrocytes. Shown in Figure 1 is the CPD data comparison between a normal case and an acute lymphoblastic leukaemia (ALL) case. Although the figure conveys a qualitative comparison, quantitative data can be calculated by considering the five light scatter angles: Axial Light Loss (ALL) A°, Low Angle Light Scatter (LALS) 5.1°, Lower Median Angle Light Scatter (LMALS) 10°−20°, Upper Median Angle Light Scatter (UMALS) 20°−42°, and a fifth scatter channel called MALS, being the sum of UMALS and LMALS regions [10].

Leukaemia and lymphoma are among the major haematological malignancies in Malaysia, comprising about 4.4% and 5.5% of the total reported cancer cases between 2007 and 2011, respectively [11]. Given the cancer burden, it is of utmost importance for patient stratification, not only in healthcare facilities in large metropolitan cities but also in rural primary and secondary healthcare facilities equipped with only basic diagnostic equipment such as FBC analysers. This present study aims to offer an insight into utilising the CPD parameters in their singular form or in a simple combined algorithm to delineate neoplastic from non-neoplastic haematological cases, followed by singular parameters that could differentiate reactive cases from neoplastic haematological cases and thereafter delineate these cases by subtype. Ultimately, the knowledge on the usefulness of the algorithm and singular parameters will be applicable for any clinical diagnostic laboratory in any part of the world that utilises any FBC analysers with light scatter angle information technology.

## 2. Materials and Methods

### 2.1. Collection of Subject’s Data and Cases

Data of 1056 apparently healthy (non-neoplastic) subjects were retrieved from a previous study to determine haematological reference intervals in Malaysian adults between September 2011 and February 2019, as described in Ambayya et al. (2014), and were retrospectively analysed [12]. Cases diagnosed with renal disease, immune thrombocytopenia, and thalassaemia were excluded. Additionally, 47 cases which were reactive due to viral and/or bacterial infection were included in the non-neoplastic group alongside normal cases, making a total of 1103 subjects in the non-neoplastic group. 

Data of 245 haematological malignancy cases diagnosed between 2015 and 2019 were retrieved for retrospective analysis in the Department of Haematology, Hospital Ampang, which serves as the national referral centre for Haematology in Malaysia. These cases consisted of 62 acute myeloid leukaemia (AML), 30 acute promyelocytic leukaemia (APL), 54 acute lymphoblastic leukaemia (ALL), 47 lymphomas, 28 chronic lymphoblastic leukaemia (CLL), 12 chronic myeloid leukaemia (CML) and 12 myelodysplastic syndrome (MDS). Two groups with small sample size (CML and MDS; 12 samples, respectively) were excluded in order to produce statistically and clinically significant findings. 

All cases were diagnosed according to the diagnostic workup of acute leukaemia guidelines from the World Health Organisation of haematology malignancies diagnosis (2016), College of American Pathologists and the American Society of Haematology, which includes morphological review, immunophenotyping using flow cytometry, molecular and cytogenetic analysis [13,14]. For all disease subsets in the neoplastic group, only newly diagnosed cases were included. Cases of treated patients who were in the following categories were excluded: remission, relapsed, refractory. 

Ethical approval for this study was obtained from Medical Research Ethics Committee of the Ministry of Health of Malaysia (Research ID 10-277-5480 and NMRR 17-2708-38327). All subjects provided written informed consent prior to data collection. Samples were collected in adherence to International Council for Standardisation in Haematology [15,16,17].

For the prospective validation cohort, 9 mL of blood samples were collected in three K2 EDTA tubes for full blood count and smears, flow cytometry immunophenotyping and molecular analysis. For cytogenetic analysis, 2 ml of bone marrow aspirates in transport medium were collected. Blood and bone marrow sample analysis was carried out as described in a previous study [12]. The FBC protocol was carried out on Unicel DxH 800 (Beckman Coulter, Miami, FL, USA) within 6 h of sample collection based on the guidelines provided by the International Council for Standardisation in Haematology (ICSH) [15]. The SP1000i automated slide maker (Sysmex, Kobe, Japan) was used to perform automated blood smears and staining for all cases. May–Grunwald–Giemsa staining was performed on the bone marrow aspirates. 

Statistical analyses were carried out using IBM SPSS Statistics version 22 software (SPSS, Chicago, IL, USA). Distribution of each parameter was tested using Kolmogorov–Smirnov test of normality. Assessment of significant differences between each of the CPD parameters was performed using one-way analysis of variance (ANOVA) with a test of homogeneity of variance. As for the parameters that fulfilled the homogeneity of variance, Tukey post hoc test was carried out, whereas for the parameters which did not fulfil the homogeneity of variance, Games–Howell post hoc test was performed. A Welch test was also performed to confirm parameters which did not meet the homogeneity of variance. Following comparison between the groups, whereby differences which yield *p*-values of <0.05 were considered as a candidate to generate ROC curves for further analysis. Within this study, significance was defined as having a *p*-value of <0.05 and AUC of >0.7 in the ROC curve analysis. Cut-off points of each of the shortlisted parameters were also determined by considering the parameters’ sensitivity and specificity.

In this study, several strategies were utilised: first, to distinguish haematological neoplastic from non-neoplastic cases in which five groups (AML, APL, ALL, lymphomas and CLL) and second, discrimination of reactive cases from the non-neoplastic cases. Two-tailed t-test was conducted to analyse the significant differences between the values generated by each leukocyte CPD parameter from both groups. CPD parameters with *p*-values of less than 0.05 were then selected for (Receiver Operating Characteristic) ROC analysis. 

Based on our previous study on establishment of reference intervals for CPD parameters, no clinically significant differences between the age group, gender and ethnicity were discovered upon performing statistical analysis, so the data was combined for these factors [18]. We adopted a similar strategy in this study, as the main aim of this study is to develop algorithms that are applicable across different haematological malignancies as a flagging system for the FBC analysers regardless of the age group, gender and ethnicity of the patients. 

For identification of leukaemia subtypes using CPD parameters, two-tailed t-test was performed upon different leukaemia subtype pairs. The significant differences of the CPD parameters between the pairs (AML versus APML, AML versus ALL, APML versus ALL and ALL versus CLL) were evaluated. Only parameters with *p* < 0.05 were selected for subsequent ROC analysis.

ROC analysis determines the diagnostic power of each parameter based on the AUC, sensitivity, and specificity. For this study, parameters with AUC > 0.9 are considered to have strong utility as a diagnostic tool. Cut-off points were also determined while maintaining good sensitivity and specificity values (>80%).

### 2.2. Development and Validation of Novel Algorithm

Novel algorithm was developed by incorporating the CPD parameters with AUC > 0.9 to differentiate neoplastic from non-neoplastic samples in this study. Validation of this algorithm was carried out in a prospective cohort of 284 cases of various disorders to ensure the algorithm is robust and is able to delineate neoplastic from non-neoplastic samples in a routine diagnostic laboratory sample processing. Cases that were included in the validation cohort consisted of 118 lymphomas, 39 AML, 32 multiple myeloma, 22 MDS, 21 ALL, 4 CLL, 5 CML, 5 PNH, 4 reactive (infection), 4 thalassemia and 29 other conditions (including anaemia, sarcoma, plasmacytoma, and other non-neoplastic disorders). This was then followed by the validation of the single CPD parameter (MN-AL2-NE) in a prospective cohort of 192 samples (154 neoplastic cases and 38 reactive cases) to assess the reliability in distinguishing neoplastic from the reactive cases. Area under the curve (AUC), sensitivity (true positive rate), specificity (false positive rate), positive predictive value (ppv) and negative predictive value (npv) of the novel algorithm were calculated using VassarStats (http://vassarstats.net/clin1.html) in these two prospective cohorts.

## 3. Results

### 3.1. Algorithm Development Using Preliminary Retrospective Chort 

A total of 1472 (1103 non-neoplastic; 245 neoplastic) cases were included in this study. The elaborated breakdown of the cases is listed in Table 1. 

The ROC analysis (as previously described) revealed four separate leukocyte CPD parameters with AUC more than 0.9 as follows: neutrophil parameters (SD-V-NE, MN-C-NE), lymphocyte parameter (MN-UMALS-LY), and monocyte parameter (SD-AL2-MO). These parameters have good diagnostic power in distinguishing neoplastic from non-neoplastic haematological cases.

Through the combination of four leukocyte parameters, a simple algorithm, [(SD-V-NE*MN-UMALS-LY*SD-AL2-MO)/MN-C-NE] was formulated to distinguish neoplastic from non-neoplastic haematological cases, which generated an excellent AUC of 0.989 with high levels of sensitivity and specificity of 95.83% and 96.26%, respectively, at a cut-off point of 106.44. The ROC of single parameters and combined algorithm are shown in Figure 2 below. The ROC data of the parameters and algorithm are listed below in Table 2. Box and whisker plots depicting this algorithm were generated for the two comparison groups (neoplastic vs. non-neoplastic) as shown in Figure 3. 

The neutrophil parameter, MN-AL2-NE was the only parameter useful in distinguishing reactive from neoplastic cases with an AUC of 0.948 at a cut-off point of 147.5, with sensitivity of 78.47% and specificity of 80.43% (Figure 4). Box and whisker plots illustrating the MN-AL2-NE parameter were generated for the neoplastic group versus the reactive group (Figure 5) and for all subtypes in neoplastic group versus the reactive group (Figure 6). 

Several CPD parameters with AUC > 0.7 were found useful for the identification of leukaemia subtypes, in which: AML can be differentiated from the other subtypes of leukaemia using neutrophil parameters of SD-MALS-NE and SD-UMALS-NE; APL by neutrophil parameter MN-V-NE and monocyte parameter SD-V-MO; ALL using neutrophil parameters of MN-MALS-NE and MN-LMALS-NE; and CLL by monocyte parameter SD-C-MO (Table 3).

A schematic representation of all the algorithms devised in this study is shown in Figure 7.

### 3.2. Algorithm Validation 

The first algorithm for neoplastic versus non-neoplastic [(SD-V-NE*MN-UMALS-LY*SD-AL2-MO)/MN-C-NE] was prospectively evaluated among 284 cases comprising of neoplastic and non-neoplastic cases as depicted in Table 4. This algorithm generated 98.50% (95% CI: 0.89, 0.96) sensitivity and 88.89% (95% CI: 0.63, 0.99) specificity in distinguishing neoplastic from non-neoplastic cases. The positive predictive accuracy to discriminate these two groups was 99.3% (95% CI: 0.97, 0.99) whereas the negative predictive accuracy was 80.0% (95% CI: 0.56, 0.93). 

The single CPD parameter MN-AL2-NE was evaluated in a prospective cohort of 192 samples (154 neoplastic cases and 38 reactive cases). The algorithm resulted in 91.6% (95% CI: 0.86, 0.95) sensitivity and 100% (95% CI: 0.88, 1.00) specificity. The positive predictive accuracy to discriminate between these two groups was 100% (95% CI: 0.97, 1.00), whereas the negative predictive accuracy was 74.51% (95% CI: 0.60, 0.85). 

## 4. Discussion

In this study we hypothesised that haematological neoplastic cases differ from the non-neoplastic cases in terms of the morphological structure of the cells, including size and internal complexity, and that the CPD parameters would be useful in distinguishing these differences based on the VCS module and five light scatter angle measurements of cells using Beckman Coulter’s Unicel DxH800. Other studies utilizing similar Beckman Coulter’s VCS technology (LH500, LH750 and DxH800) also supported this notion in exhibiting the utility of CPD parameters in diagnosing a number of haematological disorders [19,20,21,22,23,24].

A study by Yang et al. (2014) on the classification of acute leukaemia using CPD parameters resulted in a development of multi-parametric discriminating model that was able to predict the lineage of 405 newly diagnosed acute leukaemia cases. By using a 21 CPD parameter model, 40/47 ALL cases with specificity of 94.2% and sensitivity of 85.1% were identified. For APL, using 13 CPD parameter model, 10/10 cases were identified with specificity and sensitivity of 100% [22]. Yang’s CPD parameters’ model for ALL are extensive, comprising parameters of neutrophils, monocytes, eosinophils and lymphocytes. In contrast, our ALL discriminating model includes only the parameter of neutrophils. Such a comprehensive model was also presented by APL, in contrast to our APL discriminating model, which only includes the neutrophil and monocyte parameters. The differences may arise due to the distinct analysis method used by both studies. For a more reliable outcome, only CPD parameters with good diagnostic ability (AUC) were selected through ROC for identification of leukaemia subtypes in our study (Table 3). ROC analysis is not included in the study performed by Yang and team [22]. 

Virk and team studied the utility of CPD parameters to distinguish AML from normal and reactive cases. The authors identified MNV (mean neutrophil volume), MNV-SD (standard deviation of neutrophil volume), MMS (mean monocyte scatter), and MLS (mean lymphocyte scatter) as the best parameters to distinguish AML from normal control (AUC > 0.9). MNV-SD was the most significant parameter, with the highest AUC of 0.972, with sensitivity and specificity of 94% and 95%, respectively. For AML versus reactive cases, MNC-SD (standard deviation of neutrophil conductivity), MNS-SD (standard deviation of neutrophil scatter), MNV-SD, MNV, MMV-SD (standard deviation of monocyte volume), MMV (monocyte volume) and MMS-SD (standard deviation of monocyte scatter) were the best parameters to rule out reactive cases from AML cases (AUC > 0.7). In separate study by Gaspar et al., complete blood count (CBC), volume, conductivity and scatter parameters of leukocytes were compared between 103 CML cases versus 58 reactive cases and 100 pregnant women. The study identified MNV (mean neutrophil volume), MNC-SD (standard deviation of neutrophil conductivity) and MLS (mean lymphocyte scatter) as the best parameters to distinguish CML from reactive and pregnancy groups (AUC > 0.9; sensitivity and specificity: 94%).

Both Virk et al. and Gaspar et al. only focused on the comparison of CPD parameters among specific types of neoplastic cases, i.e: AML and CML [23,24]. In contrast, we performed a comparison of the CPD parameters among a broad group of neoplastic cases (Table 1). Virk et al. claimed MNV-SD was the most significant parameter to distinguish AML from non-neoplastic cases [24]. A similar result was observed in our study across all neoplastic cases, in which SD-V-NE (standard deviation of neutrophil volume) is the most significant parameter found to distinguish neoplastic from non-neoplastic cases (AUC: 0.978; sensitivity: 94%; specificity: 95%). The selected parameters to rule out reactive cases from neoplastic cases in both AML and CML studies were extensive (as previously described), including several neutrophil and monocyte parameters. Contrary to this, we identified MN-AL2-NE (neutrophil mean axial light loss) as the sole discriminating factor for ruling out reactive cases from neoplastic cases.

To the best of our knowledge, this is the first CPD comparison study which includes large cohort of neoplastic cases that comprise of different types of haematological disorder (AML, APL, ALL, chronic leukaemia, lymphoma, PNH, MM, MPD and MDS) against non-neoplastic cases (Figure 6). The detailed stepwise screening of leukaemia subtypes using CPD parameters as portrayed by this study has yet to be reported elsewhere. 

A total of 126 CPD parameters were considered as candidates to develop the neoteric algorithm in this study. The most significant parameters to screen out neoplastic from non-neoplastic cases were SD-V-NE, MN-C-NE, MN-UMALS-LY, SD-AL2-MO, with SD-V-NE as the most significant with the highest AUC of 0.978, sensitivity of 94.14% and specificity of 95.21%. Although the results were significant, we want to improve the outcome by devising algorithm equations using all the significant CPD parameters. The equation was devised for the comparison groups: neoplastic versus non-neoplastic cases. The results improved; the specificity and sensitivity of the algorithms were higher compared to the single parameters as shown in Table 2. A similar method was also implemented by Zhu et al. [25]. The authors incorporated single parameters LV (mean lymphocyte volume); LV-SD (lymphocyte volume standard deviation); LC (lymphocyte conductivity) into lymph index LV × LV-SD ÷ LC and achieved 91.67% sensitivity and 97.2% specificity for diagnosing viral infections [24].

The performance of the algorithm was evaluated in the validation set of 284 samples comprising a plethora of cases that were processed routinely (118 lymphomas, 39 AML, 32 multiple myeloma, 22 MDS, 21 ALL, 22 other cases (Idiopathic thrombocytopenic purpura, iron deficiency anaemia, anaemia of chronic disease) 6 CML, 5 PNH, 4 CLL, 4 reactive, 4 thalassemia, 3 aplastic anaemia and 3 polycythaemia rubra vera). By using the algorithm equation [(SD-V-NE*MN-UMALS-LY*SD-AL2-MO)/MN-C-NE], we identified 262 out of 266 neoplastic cases and ruled out 16 out of 20 non-neoplastic cases, with sensitivity of 98.50% and specificity of 88.89%. False negative results in neoplastic samples could be explained by low percentages of abnormal cells with low WBC counts, ranging between 0.9 to 7.9 × 10^9^ L which probably led to the erroneous exclusion of these neoplastic samples. However, we could not explain the reason on why two non-neoplastic samples, including Type II von Willebrand Disease and thalassemia with WBC counts within the normal limit, were incorrectly assigned as neoplastic after full blood picture smear review. Details of the four false negative and two false positive cases are summarised in Appendix A.

As it is equally paramount to ensure that the algorithm is robust in distinguishing neoplastic haematological cases from reactive cells, the single CPD parameter [MN-AL2-NE] was also assessed in a 192 prospective cohort alongside with the algorithm that we have developed. With the use of MN-AL2-NE as a single CPD discriminant, 141 neoplastic cases were successfully delineated from the reactive cases. Although this parameter was highly specific in differentiating neoplastic cases from the reactive cases, 13 out of 154 of the neoplastic cases were falsely grouped as reactive. This could be explained by the low percentages of abnormal cells in the neoplastic samples, which had low WBC counts, ranging between 3.7 to 11.4 × 10^9^ L, which probably led to erroneous exclusion of these neoplastic samples, as summarised in Appendix A.

As for the other CPD parameters that were studied to distinguish subtypes of leukaemia as reported in Table 3, no prospective validation was carried out as the AUC (<0.9), and sensitivity, as well as specificity of these parameters, were lower than 80%. In order to overcome this limitation, these parameters need to be tested in a larger cohort of neoplastic haematological cases in future, followed by a prospective validation study.

The main aim of this study is to distinguish the neoplastic haematological samples from the non-neoplastic samples using the CPD parameters in order to expedite the diagnostic and clinical management of patients, especially in primary health care facilities prior to performing more sophisticated and laborious testing in tertiary health care institutes. This was achieved by the algorithm we have developed, which has excellent sensitivity and specificity in distinguishing the neoplastic haematological cases from the non-neoplastic samples. In order to ensure that the reactive cases are not included erroneously as the neoplastic sample, MN-AL2-NE as the single parameter will be the second-line discriminant parameter. Our strategy has been proven to be powerful in the prospective cohort study using both discriminants, as described, with excellent positive predictive accuracy to distinguish neoplastic from non-neoplastic samples [99.3% (95% CI: 0.97, 0.99)] and neoplastic cases from the reactive cases (100% (95% CI: 0.97, 1.00)).

The clinical aim of this study is not to replace gold standards of diagnosis of haematological malignancies [26], but rather to form a flagging system which may alert laboratorians, pathologists, and physicians during the initial screening using full FBC analysers. This method is applicable for analysers that are equipped with CPD parameters that measures cell properties at various scatter angles, similar to the VCS technology developed by Beckman Coulter. There are no publications on algorithm development using CPD parameters on haematological malignancies on other FBC analysers from different manufacturers, so further investigations are required to evaluate the similarity of these parameters across various FBC analyser technologies. This will be especially beneficial in primary and secondary healthcare centres which lack the advanced diagnostic facilities and technical expertise needed to execute the prompt actions required for the timely making of clinical decisions and patient management.

## 5. Conclusions

This study highlighted the utility of leukocyte CPD parameters in distinguishing neoplastic from non-neoplastic cases and the identification of leukaemia subtypes. The neoteric algorithm and single CPD parameters explored in this study have shown excellent correlation with the gold standard of haematological malignancies diagnosis by morphologic and immunophenotypic studies. This novel algorithm was also proven diagnostically powerful in the preliminary evaluation of leukocyte morphology before the review of blood smears. The use of CPD parameters as an initial screen and alert for neoplastic haematological cases will be useful as a timely and rapid alert and flagging system for the laboratory, especially in settings with limited resources. This is essential in countries with rural primary health care facilities where the clinical management of patients relies on limited laboratory equipment in providing diagnostic information such as full blood count. Hence, these findings are promising in the elucidation of the usefulness of CPD parameters in the development of rapid and useful FBC-based diagnostic tools for neoplastic haematological disorders.

## Figures and Tables

**Figure 1 diagnostics-11-01652-f001:**
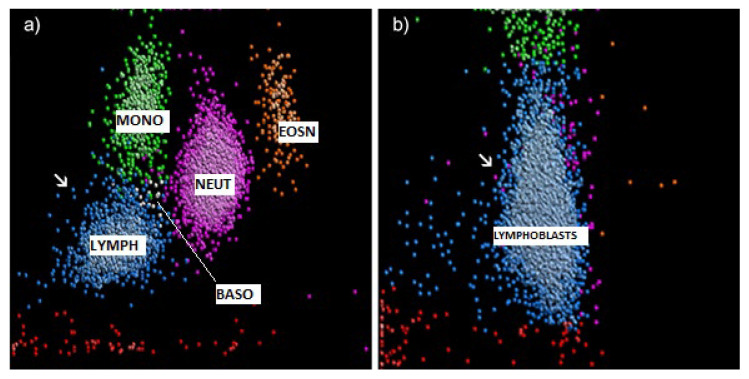
(**a**) Volume CPD presentation of a normal case. Blue population represents the lymphocytes (lymph), magenta population represents the neutrophils (neut), green population represents the monocyte, orange population represents the eosinophils and yellow population represents the basophils; (**b**) CPD (volume parameter) presentation of ALL cases. The area marked with white arrows on the blue population labelled as “lymphoblasts” shows the clearest distinction between the two cases, being the lymphoblast population in ALL cases. Appendix A provides descriptions of the CPD parameters.

**Figure 2 diagnostics-11-01652-f002:**
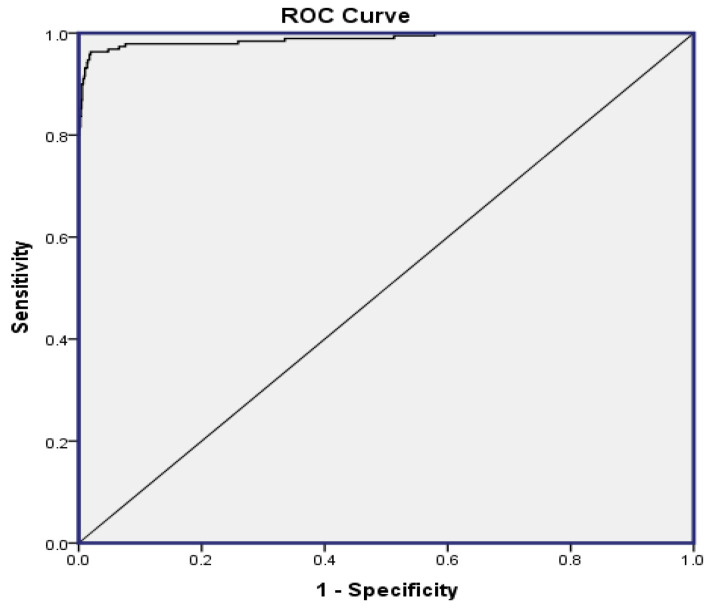
ROC curve of the novel algorithm generated to distinguish neoplastic from non-neoplastic cases: [(SD-V-NE*MN-UMALS-LY*SD-AL2-MO)/MN-C-NE] with AUC: 0.989; 95% CI: 0.980–0.998; Cut-off: 106.44; Sensitivity: 95.83%; Specificity: 96.26%.

**Figure 3 diagnostics-11-01652-f003:**
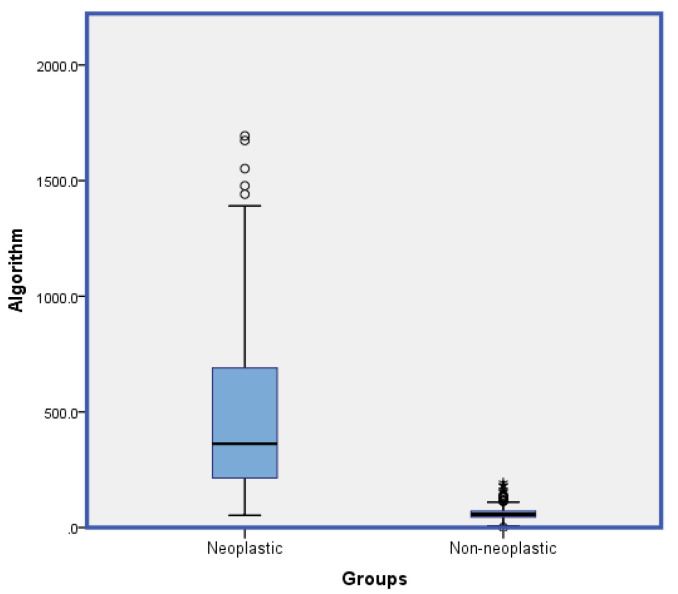
Box and whisker plot for the algorithm [(SD-V-NE*MN-UMALS-LY*SD-AL2-MO)/MN-C-NE] depicting a comparison of the two groups in this study (neoplastic vs. non-neoplastic).

**Figure 4 diagnostics-11-01652-f004:**
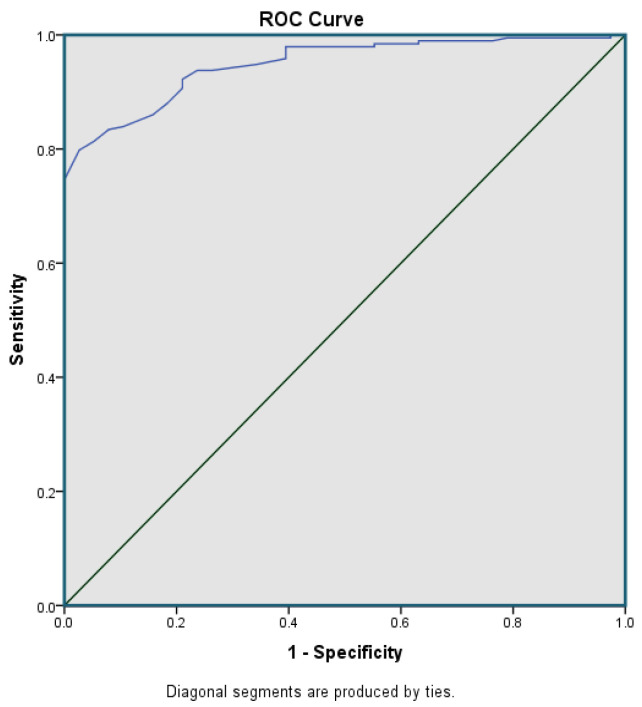
ROC curve of the single CPD parameter used to distinguish reactive cases from neoplastic cases (AUC: 0.948; 95% CI: 0.921–0.976; Cut-off: 147.5; Sensitivity: 78.47%; Specificity: 80.43%).

**Figure 5 diagnostics-11-01652-f005:**
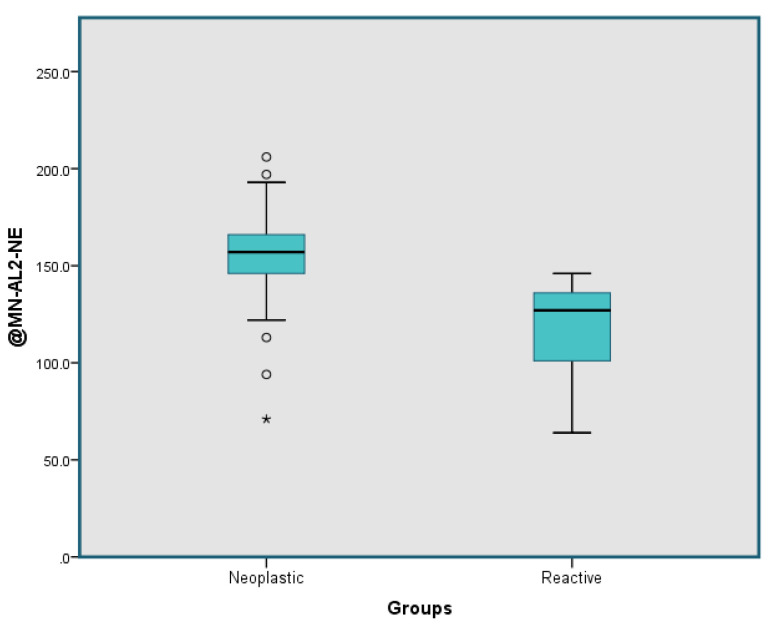
Box and whisker plot for the MN-AL2-NE parameter depicting a comparison of the two groups in this study (neoplastic vs. reactive). There was an outlier marked as “*” in the neoplastic group.

**Figure 6 diagnostics-11-01652-f006:**
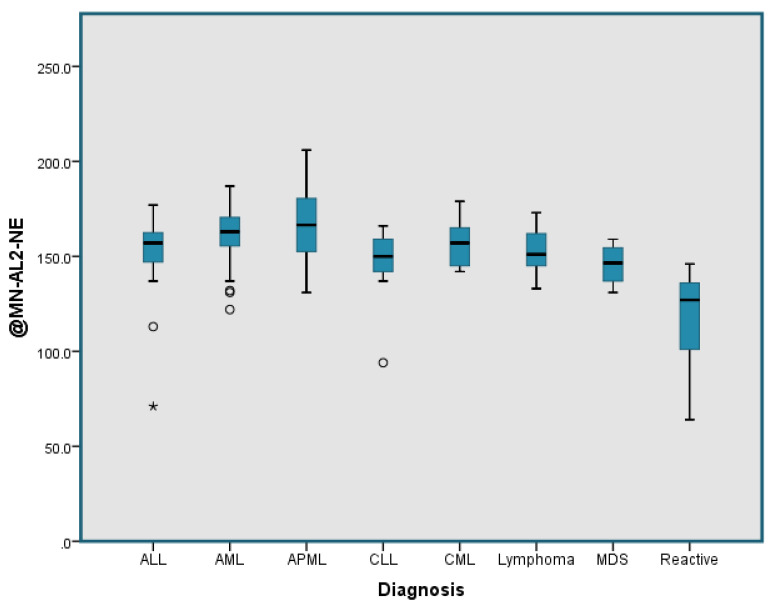
Box and whisker plot for the MN-AL2-NE parameter depicting subtypes of the neoplastic group (ALL, AML, APML, CLL, CML, lymphoma, MDS) versus the reactive group. There was an outlier marked as “*” in the ALL group.

**Figure 7 diagnostics-11-01652-f007:**
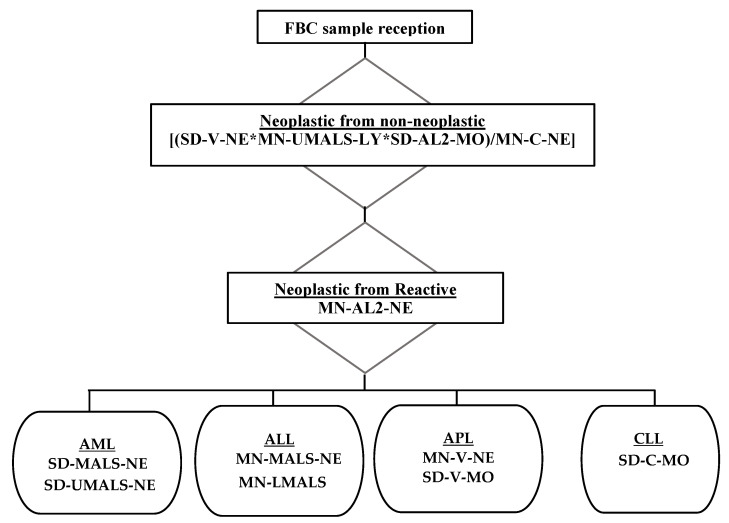
This figure depicts the algorithm devised to firstly, distinguish neoplastic cases from non-neoplastic cases and secondly, for the exclusion of reactive cases from neoplastic cases and followed by leukaemia lineage and subtypes identification.

**Table 1 diagnostics-11-01652-t001:** Distribution of cases included in this study.

Diagnosis	Cases (*n*)	Total (*n*)
Non-neoplastic		
		1103
Normal	1056	
Reactive	47	
Neoplastic		245
Acute myeloid leukaemia (AML)	62	
Acute promyelocytic leukaemia (APL)	30	
Acute lymphoblastic leukaemia (ALL)	54	
Lymphoma	47	
Chronic lymphocytic leukaemia (CLL)	28	
Chronic myeloid leukaemia (CML)	12	
Myelodysplastic syndrome (MDS)	12	

**Table 2 diagnostics-11-01652-t002:** ROC data of leukocyte CPD parameters used to distinguish neoplastic from non-neoplastic cases.

Parameters	AUC	95% CI	Cut-Off		Sensitivity (%)	Specificity (%)
			Neoplastic	Non-neoplastic		
SD-V-NE	0.978	0.967–0.989	>18.95	<18.95	95.14	95.21
MN-C-NE	0.926	0.902–0.950	<148.50	>148.50	83.60	90.30
MN-UMALS-LY	0.907	0.880–0.934	>59.50	<59.50	81.94	84.19
SD-AL2-MO	0.919	0.888–0.951	>16.20	<16.20	86.11	91.48
Algorithm	0.989	0.980–0.998	>106.44	<106.44	95.83	96.26

**Table 3 diagnostics-11-01652-t003:** The ROC data for each leukocyte CPD to distinguish subtypes of leukaemia.

Parameters	AUC	95% CI	Cut-Off	Sensitivity (%)	Specificity (%)
AML					
SD-MALS-NE	0.715	0.632, 0.798	15.81	71.43	70.10
SD-UMALS-NE	0.730	0.650, 0.811	17.70	71.43	66.36
APL					
MN-V-NE	0.723	0.601, 0.844	159.50	67.86	67.18
SD-V-MO	0.776	0.679, 0.872	33.68	78.57	77.86
ALL					
MN-MALS-NE	0.710	0.625, 0.796	129.50	68.75	62.61
MN-LMALS-NE	0.713	0.629, 0.797	122.50	68.75	63.48
CLL					
SD-C-MO	0.743	0.600, 0.886	11.10	71.43	73.79

**Table 4 diagnostics-11-01652-t004:** Validation of algorithm in distinguishing neoplastic from non-neoplastic cases.

Algorithm	Prediction	Neoplastic	Non-Neoplastic	Total
[(SD-V-NE*MN-UMALS-LY*SD-AL2-MO)/MN-C-NE	Positive	262	2	264
	Negative	4	16	20
	Total	266	18	284

## Data Availability

The data used in this study are available as Appendix A.

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
