# Peer review of "Neoteric Algorithm Using Cell Population Data (VCS Parameters) as a Rapid Screening Tool for Haematological Disorders"

_diagnostics, 2021, doi:10.3390/diagnostics11091652_

Round 1

Reviewer 1 Report

This is an excellent paper on the use of novel CPD parameters in the differential diagnosis of hematological malignancies. The authors are to be congratulated and I suggest acceptance of the paper.

Author Response

Thank you so much for your review and kind words. It has been a great joy that our work has been appreciated. God bless you. 

Reviewer 2 Report

Ambayya and Co-workers describe an original and simple algorithm on cell population data parameters. Algorithms have already been published, but there is still a need for such simple tools in the field of hematology, considering the insufficiently exploited data generated by full blood count analysers. The manuscript is well written although rare sentences would need to be reworked (namely line 160 and the last sentence in Conclusions, line 356). This work constitute an interesting tool for neoplastic screening and of interest in the light of the huge amount of cases studied. The Figures help the reader to have an overview of the data presented.

There are some points that might further improve the quality of the report :

1) The excellent positive predictive accuracies should be more highlighted in the scope of a screening tool, because it constitutes, in my opinion the main statement of the manuscript.

2) 13 out of 154 neoplastic cases were falsely grouped in the prospective validation cohort. More details are required in the text or even in a table to sum up the disease, age, sex, CPD parameters, WBC, for instance. Similarly for the 4 false negative/positive cases in the validation cohort between neoplastic and non-neoplastic samples.

3) Even if statistical analyses are not relevant, a comment on results obtained for idiopathic thrombocytopenic purpura, iron deficiency anaemia, anaemia of chronic disease, CML, PNH, thalassemia, aplastic anaemia and polycythemia vera would provide useful information for future studies.

3) The Beckman Coulter technology was used for this work, but this point is not sufficiently discussed when the authors refer to other studies, by specifying that other teams also used Beckman Coulter devices, and if other analysers could provide similar analytical performances (Sysmex and Siemens technologies could also be discussed for example).

4) This work also provides preliminary results to differentiate haematological disorders. However, this point should be further discussed and tempered considering AUC<0.9 and the absence of prospective validation cohort.

5) In Conclusions, : « This novel algorithm was also proven diagnostically powerful in the preliminary evaluation of leukocyte morphology without the review of blood smears. » is correct but the use of « before the review of blood semars » (or another suggestion to define) instead of « without the review of blood smears » would further place this screening in the context of comprehensive care before tertiary health care institutes.

Author Response

Dear Reviewer,

Thank you very much for your reviews and feedbacks to improve this manuscript. We truly appreciate all your recommendation and we have tried our best to improve this manuscript based on your comments. Thank you again. 
